# Stacked Feynman-Kac: A Generalised Method for Within-Timestep Sampling of Intermediate Distributions for Diffusion Models

## Abstract

Generative diffusion models have proven highly effective for both unconditional generation and conditional tasks. Sequential Monte Carlo (SMC) methods provide a principled framework for conditional sampling from diffusion priors, with applications including inpainting, super-resolution, and de-blurring. However, existing SMC-based approaches typically only propagate particles across timesteps and does not sufficiently explore the intermediate target distributions. To address this, we introduce the *Stacked Feynman-Kac* (SFK) algorithm, which enables sampling of intermediate distribution estimates at each timestep. We instantiate this framework with a Hamiltonian Monte Carlo proposal, which exploits gradient information to efficiently explore the intermediate targets. We further develop a computationally efficient variant of this approach and demonstrate that it achieves superior performance across a range of benchmark inverse problems and datasets.

## 1 Introduction and Related Work

Inverse problems (Daras et al., 2024) are a class of problems in which the goal is to recover a clean signal from degraded observations. More formally, given noisy, sparse, or incomplete observations $y$, we aim to recover the clean signal $x$ via

$$y = \mathcal{A}(x) + \epsilon, \tag{1}$$

where $\mathcal{A}$ is a (possibly non-linear) degradation operator and $\epsilon$ denotes observation noise. Canonical examples include inpainting, super-resolution, and de-blurring, where $\mathcal{A}$ masks, downsamples, or blurs the clean signal respectively. Such problems are inherently ill-posed, many signals $x$ may be consistent with a given observation $y$, and are naturally cast as a Bayesian inference problem, in which we seek to characterise the posterior distribution

$$p(x \mid y) \propto p(y \mid x)\, p(x), \tag{2}$$

where $p(y \mid x)$ is the likelihood induced by the observation model and $p(x)$ is a prior over clean signals. A powerful strategy for addressing this class of problems is to learn a rich prior distribution over clean data using a diffusion model $p_\theta$ (Sohl-Dickstein et al., 2015; Ho et al., 2020; Song et al., 2021a;b). Diffusion models define a forward noising process that gradually corrupts data into noise, and learn to reverse this process to generate high-quality samples, making them highly expressive priors for complex data distributions. Conditional sampling methods (Zhao et al., 2025) can then be applied to approximate the posterior $p(x \mid y)$ using this learned prior, enabling reconstruction of clean signals from corrupted observations without requiring task-specific retraining.

Various methods have been developed for conditional sampling in the context of inverse problems. Early approaches were often task-specific: Lugmayr et al. (2022) proposed a conditional sampling algorithm tailored to inpainting, while Li et al. (2022) developed a dedicated method for super-resolution. A more general class of methods instead modifies the unconditional reverse diffusion process using gradient information from the

---

$^{0\dagger}$Equal contribution.

likelihood. Dhariwal & Nichol (2021) introduced classifier guidance, which steers the reverse process using the gradient of a classifier. Building on this, Chung et al. (2023) proposed Diffusion Posterior Sampling (DPS), which approximates the intractable posterior score directly, and Kawar et al. (2022) introduced Denoising Diffusion Restoration Models (DDRM) for linear inverse problems with known degradation operators. Wang et al. (2023) further extended this line of work to noisy linear inverse problems. While effective in practice, these methods rely on approximations to the true posterior score and therefore lack theoretical guarantees on the quality of the resulting samples.

A more principled alternative is to frame conditional sampling as an inference problem and apply Sequential Monte Carlo (SMC, Del Moral et al., 2006; Chopin et al., 2020), a class of Feynman-Kac (FK) (Del Moral, 2004) algorithms, that approximates a sequence of target distributions using a population of weighted particles. At each step, particles are propagated forward via a proposal kernel, reweighted according to an incremental importance weight, and optionally resampled to concentrate computational effort on high-probability regions. SMC methods are asymptotically exact, that is as the number of particles grows, the particle approximation converges to the true target distribution, and are applicable to a broad range of conditioning criteria without requiring task-specific retraining.

In the diffusion model setting, the natural sequence of SMC targets is defined by the intermediate noised distributions $p_\theta(x_t \mid y)$ for $t = T, \ldots, 0$. Trippe et al. (2023) and Wu et al. (2023) independently proposed particle filter algorithms that operate over this sequence, with Wu et al. (2023) additionally introducing *twisting*, a technique that incorporates heuristic likelihood approximations into the proposal. Subsequent work has extended this framework to a wider class of inverse problems and diffusion architectures (Cardoso et al., 2024; Kelvinius et al., 2025; Chen et al., 2025). However, all of these methods propagate particles solely between timesteps, leaving the intermediate target distributions within each timestep underexplored. Increasing the number of denoising steps is a common way to correct for this however, this has diminishing returns which motivates the use of higher-order solving methods instead (Lu et al., 2022; Karras et al., 2022). Another option is to use a *corrector* (Song et al., 2021b) to explore within the same timestep or noise level, but this often involves using unadjusted Langevin dynamics which introduces a discretisation bias. An accept/reject step (Metropolis et al., 1953) can be incorporated to eliminate this bias, but is rarely used in the diffusion sampling literature.

This motivates a central contribution of our work: a within-timestep corrector with minimal additional computational overhead that targets the intermediate distributions without introducing discretisation bias. We show that Sequential Monte Carlo can be used not only to propagate samples between timesteps, but also to perform this within-timestep correction.

Markov chain Monte Carlo (MCMC) methods (Metropolis et al., 1953) provide a principled approach for drawing samples from complex, intractable distributions, and offer a natural tool for improving these within-timestep estimates, as they have been used to great effect in approximating static distributions. A particularly effective variant is Hamiltonian Monte Carlo (HMC) (Neal et al., 2011; Hoffman & Gelman, 2014), which leverages local gradient information to generate proposals that explore the target distribution far more efficiently than traditional random-walk methods. HMC has also been used as a proposal distribution within the SMC framework (Buchholz et al., 2021b; Devlin et al., 2024), demonstrating its compatibility with particle-based inference. Previous work has explored the use of MCMC within generative diffusion contexts (Dang et al., 2025; Janati et al., 2025; Corenflos et al., 2025; Kalaivanan et al., 2025; Huang et al., 2024; Wu et al., 2024); however, a key challenge is that MCMC methods typically require a Metropolis-Hastings (MH) accept-reject step to ensure asymptotic correctness, which in turn requires evaluation of the likelihood. This is non-trivial in the diffusion setting, where models are generally trained to output the score, the gradient of the time-marginal distribution, rather than the likelihood itself. Du et al. (2023) proposed energy-based diffusion models (EDMs) that instead learn the unnormalised marginal directly, and also demonstrated using unadjusted HMC dynamics as a means of sampling from traditional score-trained diffusion models. The latter is a direction we build upon in this work.

**Contributions**

Building on these foundations, we propose a framework that addresses the key limitation of existing SMC methods by enabling principled refinement of intermediate target distributions within each timestep. Our specific contributions are as follows:

1. We propose the *Stacked Feynman-Kac* (SFK) algorithm, which provides a mathematical framework for refining estimates of marginal distributions within each timestep when sampling from diffusion models.

2. We show how a Hamiltonian Monte Carlo proposal can be used to explore these intermediate marginal distributions. This HMC proposal admits a weight update that is trivial to evaluate, forgoing the traditional Metropolis-Hastings accept-reject ratio.

3. We demonstrate on benchmark inverse problems and datasets that a computationally efficient variant of our algorithm, the *Reduced Stacked Feynman-Kac* (rSFK), achieves superior reconstruction performance with minimal additional computation relative to existing SMC-based methods.

## 2 Background

### 2.1 Diffusion Models

Diffusion models (Sohl-Dickstein et al., 2015; Ho et al., 2020; Song et al., 2021b) are a class of generative models that define a *forward process* which gradually corrupts data $x_0 \sim p(x_0)$ by adding Gaussian noise over $T$ timesteps, producing a sequence of increasingly noisy latents $x_1, x_2, \ldots, x_T$. Throughout, we adopt the convention that larger $t$ corresponds to noisier latents, with the reverse process running from $t = T$ to $t = 0$ via transitions of the form $p(x_t \mid x_{t+1})$. The forward process is a fixed Markov chain defined by

$$q(x_t \mid x_{t-1}) = \mathcal{N}\left(x_t; \ \sqrt{1 - \beta_t}\, x_{t-1}, \ \beta_t \mathbf{I}\right), \tag{3}$$

where $\{\beta_t\}_{t=1}^T$ is a noise schedule chosen such that $q(x_T \mid x_0) \approx \mathcal{N}(0, \mathbf{I})$, so that the forward process destroys all structure in the data.

Generation is achieved by learning to *reverse* this process. The true reverse conditionals $q(x_t \mid x_{t+1})$ are intractable as they require marginalizing over the data distribution $p(x_0)$; conditioning on $x_0$, however, yields a tractable Gaussian,

$$q(x_t \mid x_{t+1}, x_0) = \mathcal{N}\left(x_t; \ \tilde{\mu}_t(x_{t+1}, x_0), \ \tilde{\beta}_t \mathbf{I}\right), \tag{4}$$

with posterior mean and variance given by

$$\tilde{\mu}_t(x_{t+1}, x_0) = \frac{\sqrt{\bar{\alpha}_t}\, \beta_{t+1}}{1 - \bar{\alpha}_{t+1}}\, x_0 + \frac{\sqrt{\alpha_{t+1}}(1 - \bar{\alpha}_t)}{1 - \bar{\alpha}_{t+1}}\, x_{t+1}, \qquad \tilde{\beta}_t = \frac{1 - \bar{\alpha}_t}{1 - \bar{\alpha}_{t+1}}\, \beta_{t+1}, \tag{5}$$

where $\alpha_t = 1 - \beta_t$ and $\bar{\alpha}_t = \prod_{s=1}^{t} \alpha_s$. A neural network $\epsilon_\theta$ is trained to predict the noise $\epsilon \sim \mathcal{N}(0, \mathbf{I})$ added at each step, from which a prediction of the clean sample can be obtained as

$$\hat{x}_0 = \frac{x_{t+1} - \sqrt{1 - \bar{\alpha}_{t+1}}\, \epsilon_\theta(x_{t+1}, t+1)}{\sqrt{\bar{\alpha}_{t+1}}}. \tag{6}$$

Substituting this plug-in estimate into the tractable posterior yields the learned reverse transition

$$p_\theta(x_t \mid x_{t+1}) = q(x_t \mid x_{t+1}, \hat{x}_0) = \mathcal{N}\left(x_t; \ \tilde{\mu}_t(x_{t+1}, \hat{x}_0), \ \tilde{\beta}_t \mathbf{I}\right), \tag{7}$$

which approximates the true (intractable) reverse conditional. The full reverse process is then a Markov chain

$$p_\theta(x_{0:T}) = p(x_T) \prod_{t=0}^{T-1} p_\theta(x_t \mid x_{t+1}), \tag{8}$$

with $p(x_T) = \mathcal{N}(0, \mathbf{I})$. New samples are drawn by first sampling $x_T \sim p(x_T)$ and then iteratively applying the learned reverse transitions.

## 2.2 Conditional Sampling

In many applications we wish to sample from the posterior distribution $p(x_0 \mid y)$, where $y$ is some observed measurement or conditioning signal. Assuming that $y$ depends only on the clean sample, $p(y \mid x_{0:T}) = p(y \mid x_0)$, Bayes' rule gives

$$p(x_0 \mid y) \propto p(y \mid x_0)\, p_\theta(x_0). \tag{9}$$

Direct sampling from this posterior is intractable because the prior $p_\theta(x_0)$ is available only implicitly, through the reverse Markov chain. A common workaround is to define an extended target over the full trajectory,

$$p_\theta(x_{0:T} \mid y) = \frac{p_\theta(x_{0:T},\, y)}{p(y)} = p(x_T) \prod_{t=0}^{T-1} p_\theta(x_t \mid x_{t+1}) \times \frac{p(y \mid x_0)}{p(y)}, \tag{10}$$

whose marginal at $t = 0$ is the desired conditional distribution.

**Notation.** Throughout this section we have followed the standard convention of denoting the learned reverse transitions and prior as $p_\theta(x_t \mid x_{t+1})$ and $p_\theta(x_0)$ respectively, where $\theta$ are the parameters of $\epsilon_\theta$. In the remainder of this work we drop the subscript and write $p(x_t \mid x_{t+1})$, $p(x_0)$ for brevity.

## 2.3 SMC and Feynman–Kac

Sequential Monte Carlo (SMC) (Del Moral et al., 2006; Chopin et al., 2020; Naesseth et al., 2019) is a family of algorithms that approximates a target distribution by a population of weighted particles $\{(x_{0:T}^{(i)}, w_T^{(i)})\}_{i=1}^N$. Particles are evolved through a sequence of intermediate distributions, each easier to sample from than the final target, making SMC well-suited to high-dimensional problems where direct sampling is infeasible. At each step, particles are propagated forward via a *proposal* distribution $r_t$ and reweighted by a non-negative *potential* $G_t$, which acts as an incremental importance weight; a *resampling* step is applied periodically to focus computation on high-weight particles.

The *Feynman–Kac* (FK) framework (Del Moral, 2004) provides a unifying formulation: it specifies the precise distribution that the weighted-particle system approximates as

$$\nu_T(x_{0:T}) = \frac{1}{\mathcal{L}_T} \left[ r_T(x_T) \prod_{t=0}^{T-1} r_t(x_t \mid x_{t+1}) \right] \times \left[ G_T(x_T) \prod_{t=0}^{T-1} G_t(x_t, x_{t+1}) \right], \tag{11}$$

where $\mathcal{L}_T$ is the (generally intractable) normalizing constant. When $G_t \equiv 1$, this reduces to the proposal distribution itself and SMC degenerates to plain importance sampling; the role of the $G_t$ is to *tilt* the proposal toward the target. Different choices of $\{r_t\}$ and $\{G_t\}$ give rise to different algorithms, and a central question in any SMC method for diffusion models is therefore how to choose these factors so that (i) the marginal of $\nu_T()$ at $t = 0$ matches the desired posterior $p(x_0 \mid y)$, and (ii) the resulting weights have low variance.

## 2.4 Twisted Diffusion Sampler (TDS)

The twisted diffusion sampler (Wu et al., 2023; Whiteley & Lee, 2014) is an SMC method for conditional sampling from diffusion models, built around a family of *twisted likelihoods* $\tilde{p}(y \mid x_t)$. These are tractable approximations to the intractable intermediate likelihoods $p(y \mid x_t) = \mathbb{E}_{x_0 \sim p(x_0 \mid x_t)}[p(y \mid x_0)]$, typically obtained by replacing the expectation with a plug-in estimate, e.g. $\tilde{p}(y \mid x_t) = p(y \mid \hat{x}_0(x_t))$, where $\hat{x}_0(x_t)$ is the model's $x_0$-prediction at noise level $t$. The corresponding twisted reverse transition $\tilde{p}(x_t \mid x_{t+1}, y)$ approximates the true posterior reverse transition, and is usually constructed by perturbing $p(x_t \mid x_{t+1})$ with the gradient $\nabla_{x_{t+1}} \log \tilde{p}(y \mid x_{t+1})$, in the spirit of classifier guidance.

TDS uses the twisted reverse transitions as proposals,

$$r_t(x_t \mid x_{t+1}) = \tilde{p}(x_t \mid x_{t+1}, y), \tag{12}$$

together with the weighting functions

$$G_t(x_t, x_{t+1}) = \frac{\tilde{p}(y \mid x_t)}{\tilde{p}(y \mid x_{t+1})} \cdot \frac{p(x_t \mid x_{t+1})}{\tilde{p}(x_t \mid x_{t+1}, y)}. \tag{13}$$

At the initial time-point $T$ the proposal and weighting functions are

$$r_T(x_T) = p(x_T), \qquad G_T(x_T) = \tilde{p}(y \mid x_T). \tag{14}$$

To verify correctness, we substitute (12)–(14) into the FK measure (11). The proposal ratios cancel, and the twisted likelihood ratios telescope along the trajectory:

$$r_T G_T \prod_{t=0}^{T-1} r_t G_t = p(x_T)\tilde{p}(y \mid x_T) \prod_{t=0}^{T-1} \tilde{p}(x_t \mid x_{t+1}, y) \cdot \frac{\tilde{p}(y \mid x_t)}{\tilde{p}(y \mid x_{t+1})} \cdot \frac{p(x_t \mid x_{t+1})}{\tilde{p}(x_t \mid x_{t+1}, y)}$$

$$= p(x_T) \prod_{t=0}^{T-1} p(x_t \mid x_{t+1}) \cdot \tilde{p}(y \mid x_0), \tag{15}$$

where we have used the boundary condition $\tilde{p}(y \mid x_0) = p(y \mid x_0)$. This is precisely the unnormalized joint in (10), so the FK target $\nu_T()$ coincides with $p_\theta(x_{0:T} \mid y)$. Marginalizing over $x_{1:T}$ – which the SMC algorithm does implicitly, as only the $x_0$-component of each particle is retained – therefore yields samples from $p(x_0 \mid y)$.

## 2.5 Unadjusted HMC as a proposal for SMC

Traditionally, an Euler-Maruyama (EM) (Kloeden & Platen, 1992) discretisation method is used as the proposal for the TDS framework to propagate samples between timesteps. We employ this proposal as well, but we show in Section 3.1 that our new method admits the use of other types of proposals, given certain criteria are met. In particular, we focus on the use of Hamiltonain Monte Carlo (HMC) dynamics. Therefore, the following section details the use of these dynamics in an SMC setting.

HMC (Duane et al., 1987; Neal et al., 2011; Betancourt, 2017) is an MCMC method that samples from a target density $\pi(x)$ by augmenting the state with an auxiliary momentum variable $v \sim \mathcal{N}(0, M)$, where $M$ is the so-called *mass matrix*, and simulating Hamiltonian dynamics on the joint distribution $\pi(x)\,\mathcal{N}(v; 0, M)$. In practice the dynamics are discretized with the *leapfrog* integrator, which is both volume-preserving and time-reversible – two properties that we will rely on heavily below. A single leapfrog step of size $h$ maps $(x, v) \mapsto (x', v')$ via

$$v_{\frac{1}{2}} = v + \tfrac{h}{2}\nabla \log \pi(x), \tag{16}$$

$$x' = x + h\, v_{\frac{1}{2}}, \tag{17}$$

$$v' = v_{\frac{1}{2}} + \tfrac{h}{2}\nabla \log \pi(x'), \tag{18}$$

where $\pi$ is the probability density function of the target distribution. Equations equation 16–18 are typically iterated $L$ times, with $L$ known as the number of leapfrog steps. For ease of exposition we present the derivation below for $L = 1$; the argument extends to any $L \geq 1$ unchanged. We denote the position component of the leapfrog map by $f_{LF}(x, v) = x'$.

In standard HMC, invariance of the target is preserved by accepting the proposal with the Metropolis–Hastings (MH) (Metropolis et al., 1953) probability

$$\alpha(x, x') = \min\left(1, \frac{\pi(x')\, q(x \mid x')}{\pi(x)\, q(x' \mid x)}\right). \tag{19}$$

HMC has previously been used as a proposal mechanism within SMC samplers (Buchholz et al., 2021a; Devlin et al., 2024). Devlin et al. (2024) showed that in this SMC setting the MH correction is no longer required: invariance via the MH citerion is replaced by importance reweighting, and any forward kernel $q(x' \mid x)$ may be combined with a backward kernel $L(x \mid x')$ to give the incremental weight

$$G(x', x) = \frac{\pi(x')}{\pi(x)}\, \frac{L(x \mid x')}{q(x' \mid x)}. \tag{20}$$

The freedom in choosing $L$ is non-trivial: it must admit a tractable, point-evaluable density so that equation 20 can actually be computed.

Following Devlin et al. (2024), we exploit reversibility of leapfrog (LF) to obtain such a kernel via a change of variables in the momentum. Since $x' = f_{LF}(x, v)$ is a deterministic function of $v \sim \mathcal{N}(0, M)$ once $x$ is fixed, the proposal density is

$$q(x' \mid x) = \mathcal{N}(v; 0, M) \, |J(x, v)|^{-1}, \qquad J(x, v) := \frac{\partial f_{LF}(x, v)}{\partial v}. \tag{21}$$

We choose the backward kernel to be the leapfrog map run from $(x', -v')$, which by reversibility recovers $x$ in position. The same change-of-variables argument then gives

$$L(x \mid x') = \mathcal{N}(-v'; 0, M) \, |J(x', -v')|^{-1}. \tag{22}$$

Two properties of leapfrog now combine to make the weight tractable. By reversibility, $f_{LF}(x', -v') = x$ with returned momentum $-v$. This implies

$$|J(x, v)| = |J(x', -v')|, \tag{23}$$

i.e. the forward and backward Jacobians coincide. Substituting equation 22 into equation 20 and cancelling the Jacobian terms yields the closed-form weight

$$G(x', x) = \frac{\pi(x')}{\pi(x)} \frac{\mathcal{N}(-v'; 0, M)}{\mathcal{N}(v; 0, M)}, \tag{24}$$

which can be evaluated directly from the simulated momenta. We refer to the unadjusted leapfrog proposal as *uHMC*. unadjusted MCMC dynamics have been explored in previous work Welling & Teh (2011); Chen et al. (2014); Durmus & Moulines (2017) where they sacrifice asymptotic correctness for computational efficiency. In our SMC setting, this trade-off does not arise: the importance weight in equation 24 restores asymptotic exactness regardless of any bias in the leapfrog discretization.

## 3 Methodology

### 3.1 Stacked Feynman-Kac

One important part of the problem of sampling from $p(x_0 \mid y)$ is that we have a lot of freedom in our choices of intermediate distributions, the decomposition in equation 10 is true but we could also add in additional steps into the target and have a correct marginal distribution. One such choice would be to add extra steps within the model and the FK formulation, basically just increasing the number of iterations $T$, but it has been shown before that increasing this number has diminishing returns (see Karras et al., 2022, Figure 2).

An alternative approach is to modify the FK formulation. If we can sample from a Markov kernel $\kappa(\cdot \mid x_t, y)$ that admits $p(x_t \mid y)$ as a stationary distribution. We can then use this kernel to revitalize the samples. To do this, we introduce the notation $x_t^K$ as the random variable after this extra simulation step and let $x_t^0$ be the random variable $x_t$. If we add in these refinement steps in between each iteration of the original FK formulation we get

$$\nu_T^\kappa(x_{0:T}^0, x_{1:T}^K) = \frac{1}{\mathcal{L}_T^\kappa} \left[ r_T(x_T^0) \prod_{t=0}^{T-1} r_t(x_t^0 \mid x_{t+1}^K)\kappa(x_{t+1}^K \mid x_{t+1}^0, y) \right] \times \left[ G_T(x_T^0) \prod_{t=0}^{T-1} G_t(x_t^0, x_{t+1}^K) \right], \tag{25}$$

where $\mathcal{L}_T^\kappa$ is the corresponding likelihood. In practice, we are not able to sample directly from $\kappa$ and will instead use another SMC approach to sample from this kernel. That is we will introduce the proposal kernels $r_t^\kappa$ and corresponding weighting functions $G_t^\kappa$. Inspired by the TDS and using the uHMC as a proposal in section 2.5 we let the proposal be

$$r_t^\kappa(x_t^K \mid x_t^0, y) = \tilde{p}(x_t^K \mid x_t^0, y), \tag{26}$$

where this should be some distribution that you can sample from and evaluate, using this we get the associated weighting function

$$G_t^\kappa(x_t^K, x_t^0) = \frac{\kappa(x_t^0 \mid x_t^K, y)}{\tilde{p}(x_t^K \mid x_t^0, y)}, \tag{27}$$

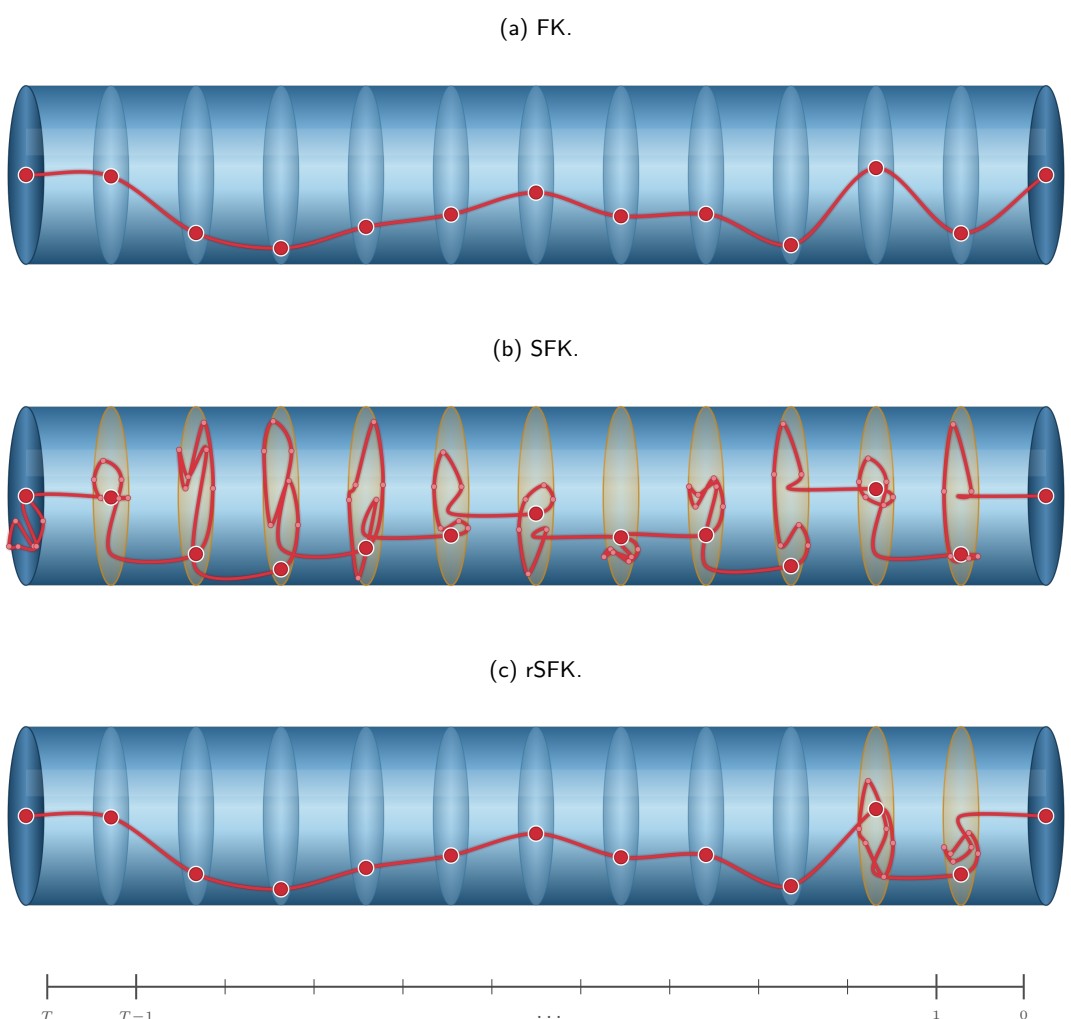

Figure 1: Visual aid for describing the differences between the three discussed sampling approaches. Each intermediate radius represents the distribution at a given timestep. FK samples dynamically over timesteps, while SFK searches within the same timestep for a better approximation of the distribution. rSFK reduces computational overhead by only using HMC near the final timestep(s).

which requires us to evaluate the $\kappa$ kernel, which might not be feasible. Following the ideas of using the uHMC as a proposal kernel (Devlin et al., 2024) we introduce the backwards kernel, through the detailed balance equation,

$$L(x_t^0|x_t^K, y) = \frac{p(x_t^0 \mid y)}{p(x_t^s \mid y)} \kappa(x_t^K \mid x_t^0, y), \tag{28}$$

of $\kappa$, by plugging this into the weighting function in equation 27 we get the new weighting function

$$G_t^\kappa(x_t^K, x_t^0) = \frac{p(x_t^s \mid y)}{p(x_t^0 \mid y)} \frac{L(x_t^0 \mid x_t^K, y)}{\widetilde{p}(x_t^K \mid x_t^0, y)}. \tag{29}$$

Using these proposals and weighting functions, we can get a double FK formulation,

$$\nu_T^\kappa(x_{0:T}^0, x_{1:T}^K) = \frac{1}{\mathcal{L}_T^K}\left[r_T(x_T^0)\prod_{t=0}^{T-1}r_t(x_t^0 \mid x_{t+1}^K)r_{t+1}^\kappa(x_{t+1}^K \mid x_{t+1}^0, y)\right]$$

$$\times \left[G_T(x_T^0)\prod_{t=0}^{T-1}G_t(x_t^0, x_{t+1}^K)G_{t+1}^\kappa(x_{t+1}^K, x_{t+1}^0)\right]. \tag{30}$$

Using this gives us a correct algorithm that generates samples from the target distribution. We formalise this fact in the following theorem.

**Theorem 1.** *The $x_0^0$ marginal of the double Feynman-Kac formulation above, $\int \nu_T^\kappa(x_{0:T}^0, x_{1:T}^K)\mathrm{d}x_{1:T}^0\mathrm{d}x_{1:T}^K$, when using the proposals and weighting functions in equations 12–14 and equations equation 26 and 29 is the target distribution $p(x_0^0 \mid y)$.*

*Proof.* We will prove this using an induction type argumentation, that is we will marginalise one variable at a time and show that after each variable we get back a form that we can recognise and the following integrals becomes trivially the same. We begin this by marginalising out the variable $x_T^0$, the terms from the Feynman-Kac formulation containing this term is, after simplification,

$$r_T(x_T^0)r_T^\kappa(x_T^K \mid x_T^0)G_T(x_T^0)G_T^\kappa(x_T^K, x_T^0) = p(x_T^0)\kappa(x_T^K \mid x_T^0, y)\tilde{p}(y \mid x_T^0),$$

taking the integral now over $x_T^0$ gives

$$\int p(x_T^0)\kappa(x_T^K \mid x_T^0, y)p(y \mid x_T^0)\mathrm{d}x_T^0 = p(y \mid x_T^K)p(x_T^K),$$

where we use that $\kappa$ has $p(x \mid y)$ as stationary distribution and that

$$p(x_T^0)p(y \mid x_T^0) = p(x_T^0 \mid y)p(y),$$

and then switching back. Given this, we can add the rest of the terms including $x_T^K$ and get

$$p(y \mid x_T^K)p(x_T^K) \times r_{T-1}(x_{T-1}^0 \mid x_T^s)G_{T-1}(x_{T-1}^0, x_T^K)$$

$$= p(y \mid x_T^K)p(x_T^K)\frac{p(y \mid x_{T-1}^0)}{p(y \mid x_T^K)}p(x_{T-1}^0 \mid x_T^K) = p(y \mid x_{T-1}^0)p(x_{T-1}^0 \mid x_T^K)p(x_T^K)p(y \mid x_{T-1}^0).$$

Taking the integral over $x_T^K$ we get,

$$\int p(y \mid x_{T-1}^0)p(x_{T-1}^0 \mid x_T^K)p(x_T^s)p(y \mid x_{T-1}^0)\mathrm{d}x_{T-1}^K = p(y \mid x_{T-1}^0)p(x_{T-1}^0), \tag{31}$$

which moved the stacked FK formulation to $\nu_{T-1}^\kappa(x_{0:T-1}^0, x_{1:T-1}^K)$. Repeating this for the rest of the variables we get the final result that,

$$\int \nu_T^\kappa(x_{0:T}^0, x_{1:T}^K)\mathrm{d}x_{1:T}^0\mathrm{d}x_{1:T}^K \propto \tilde{p}(y \mid x_0^0)p(x_0^0) \propto p(x_0^0 \mid y), \tag{32}$$

which is the target distribution. □

Noticing that we can add one of these intermediate steps, nothing stops us from adding multiple such intermediate steps in-between each time update. That is for each fixed time $t$ we have the sequence of samples $\{x_t^i\}_{i=0}^K$ where each of these samples are generated by additional applications of $\kappa$ again, in practice targeting it through the proposal and weighting function equations 26 and 29. This gives us the *stacked Feynman-Kac* (SFK) formulation,

$$\nu_T^\kappa(\{x_{0:T}^i\}_{i=0}^K) = \frac{1}{\mathcal{L}_{T,K}^\kappa}\left[r_T(x_T^0)\prod_{t=0}^{T-1}r_t(x_t^0 \mid x_{t+1}^K)\prod_{i=0}^{K-1}r_{t+1}^\kappa(x_{t+1}^{i+1} \mid x_{t+1}^i, y)\right]$$

$$\times \left[G_T(x_T^T)\prod_{t=0}^{T-1}G_t(x_t^0, x_{t+1}^K)\prod_{i=0}^{K-1}G_{t+1}^\kappa(x_{t+1}^{i+1}, x_{t+1}^i)\right], \tag{33}$$

where the $r_t$ kernels move the samples through the diffusion model and the $r_t^\kappa$ kernels explore the intermediate target distributions improving the samples. Again, following the marginalization over all intermediate targets the marginal distribution is unchanged and we still target the correct distribution. Check the Appendix for details. We provide the psuedocode for this in Algorithm 1.

### 3.1.1 Reduced Stacked Feynman-Kac

It turns out that using the above stacked Feynman-Kac formulation in equation 33 allows us to improve the generated samples but at the cost of a lot of extra computations. For each time-step we need to generate $K$ additional samples which can be quite costly.

As it is written in equation 33 we generate the same amount of intermediate steps for each time-point, this is of course not necessary and we can let the number of intermediate steps $K$ vary with $t$. In general we can introduce a sequence of non-negative integers $\{K_t\}_{t=1}^T$, where each $K_t$ is the number of intermediate steps in the stacked Feynman-Kac formulation. Note that setting $s_t = 0$ reduces it down to the standard Feynman-Kac. Since we have this freedom in choosing how many intermediate steps we are going to do, one way to reduce the number of computations needed is to only do these intermediate steps at one, or possibly a few, of the time-steps.

As an example, if we only do the intermediate steps at one time-point $t_\star$ then we set $K_t = 0$ for $t \neq t_\star$ and $K_{t_\star} > 0$. We call this version where we only do the intermediate sampling at one time-point the *reduced stacked Feynman-Kac* (rSFK) formulation.

An overview of these different versions of the Feynman-Kac formulation can be seen in figure 1 where the top corresponds to the standard Feynman-Kac, middle is the stacked Feynman-Kac and the bottom is the reduced stacked Feynman-Kac formulation. The effect of the intermediate steps is illustrated in these plots by moving the sample, exploring the distribution, at each time-point before moving to the next.

## 3.2 Using HMC for sampling intermediate steps

There are many strategies to samples from $r_{t+1}^\kappa(x_{t+1}^{i+1} \mid x_{t+1}^i, y)$ and evaluate the corresponding weight $G_{t+1}^\kappa(x_{t+1}^{i+1}, x_{t+1}^i)$. Inspired by Devlin et al. (2024) we choose to use a uHMC proposal which gives rise to a weight update that is easy to evaluate. We describe how we adapt the method for our case here, and discuss in §3.3 the practical considerations – chiefly compute and the tuning of guidance strength parameter $\lambda$ – that motivate applying it only at a small number of chosen noise levels.

Fix a noise level $t$. We propose new positions via leapfrog dynamics whose kicks use the guided gradient

$$g_\lambda(x; t, y) = s_\theta(x, t) + \lambda \nabla_x \log \tilde{p}(x_t \mid y), \tag{34}$$

where $s_\theta(x, t) = -\epsilon_\theta(x, t)/\sqrt{1 - \bar{\alpha}_t}$ is the marginal score returned by the unconditional diffusion model via Tweedie's identity, and $\lambda \geq 0$ is a tunable scale that controls the strength of the gradient guidance from the twisted likelihood. The default $\lambda = 1$ recovers the unmodified twisted-likelihood gradient; in practice we find $\lambda > 1$ improves guidance strength at intermediate noise levels. A single leapfrog step of size $h$ is then

$$v_{1/2} = v + \tfrac{h}{2} g_\lambda(x; t, y), \tag{35}$$

$$x' = x + h M^{-1} v_{1/2}, \tag{36}$$

$$v' = v_{1/2} + \tfrac{h}{2} g_\lambda(x'; t, y). \tag{37}$$

We iterate equation 35–equation 37 for $L$ steps and denote the resulting position map by $f_{LF}(x, v; t, y)$, treating $t$ and $y$ as fixed parameters.

We now show that the change-of-variables identity from §2.5 extends to this setting, despite the leapfrog kicks no longer corresponding to the gradient of any specific potential.

**Lemma 1** (Conditional change-of-variables). *Let $f_{LF}(\cdot, \cdot; t, y)$ denote the leapfrog map defined by equation 35– equation 37, and write $x' = f_{LF}(x, v; t, y)$ with $v'$ the corresponding output momentum. Assume the momentum*

*is refreshed according to $v \sim \mathcal{N}(0, M)$ independently of $(x, y)$, and define $J(x, v; t, y) := \partial f_{LF}(x, v; t, y)/\partial v$. Then the forward and backward kernel densities are*

$$\tilde{p}(x' \mid x, y) = \mathcal{N}(v; 0, M) \, |J(x, v; t, y)|^{-1}, \tag{38}$$

$$L(x \mid x', y) = \mathcal{N}(-v'; 0, M) \, |J(x', -v'; t, y)|^{-1}, \tag{39}$$

*the Jacobians coincide, $|J(x, v; t, y)| = |J(x', -v'; t, y)|$, and the SMC weight equation 20 with target factor $\tilde{p}(x \mid y)$ reduces to*

$$G^{\kappa}(x', x; y) = \frac{p(x' \mid y)}{p(x \mid y)} \cdot \frac{\mathcal{N}(-v'; 0, M)}{\mathcal{N}(v; 0, M)}. \tag{40}$$

*Proof.* For fixed $(x, t, y)$ the position output $x' = f_{LF}(x, v; t, y)$ is a deterministic function of $v$. The standard change-of-variables formula gives

$$\tilde{p}(x' \mid x, y) = q(v \mid x, y) \, |J(x, v; t, y)|^{-1}, \tag{41}$$

and the assumed independence of $v$ from $(x, y)$ reduces the conditional density to $q(v) = \mathcal{N}(v; 0, M)$, which establishes equation 38.

The leapfrog scheme equation 35–equation 37 is the composition of three shears in $(x, v)$ – a momentum kick at fixed $x$, a position drift at fixed $v$, and a second momentum kick at fixed $x'$ – each of which has unit Jacobian determinant. The full map $(x, v) \mapsto (x', v')$ is therefore volume-preserving for any vector field used in the kicks; in particular, $g_\lambda$ need not arise as the gradient of a scalar potential. Direct computation yields the time-reversal property $f_{LF}(x', -v'; t, y) = x$ in position with returned momentum $-v$, so the same change-of-variables argument applied to the reversed map establishes equation 39. Volume preservation together with reversibility implies that the position-from-momentum Jacobians at $(x, v)$ and $(x', -v')$ have equal magnitude (Devlin et al., 2024), and equation 40 follows by substituting equation 38–equation 39 into the SMC weight equation 20 and cancelling Jacobian terms. $\qquad\square$

Iterating equation 40 for $K$ steps at the same noise level, the intermediate likelihood factors telescope:

$$\prod_{i=0}^{K-1} G^{\kappa}(x^{i+1}, x^i; y) = \frac{p(x^K \mid y)}{p(x^0 \mid y)} \cdot \prod_{i=0}^{K-1} \frac{\mathcal{N}(-v'_i; 0, M)}{\mathcal{N}(v_i; 0, M)}. \tag{42}$$

**Role of $\lambda$.** The guidance scale $\lambda$ appears in the leapfrog gradient equation 34 but *not* in the SMC weight equation 40. It therefore controls the dynamics – how aggressively particles are pushed toward observation-consistent configurations during the leapfrog kicks – without altering the implicit SMC target. Larger $\lambda$ yields proposals that are more strongly guided by $y$, but at the cost of dynamics that are not stationary for the implicit target and that may overshoot high-likelihood regions; the SMC weight equation 40 corrects for this discrepancy via importance reweighting. The choice of $\lambda$ thus trades off mixing speed against weight variance, and is in general a function of the noise level at which refinement is applied.

**Remark (independence of momentum refresh).** The independence assumption $q(v \mid x, y) = q(v)$ in Lemma 1 is what makes the extension immediate. Schemes that adapt the mass matrix to $y$ would break this identity and require re-deriving the weight ratio.

**Remark (evaluating $p(x \mid y)$).** In general we don't have access to $p(x \mid y)$ and have to represent it with some approximation. We will use the same approximation that we do for $\tilde{p}(y \mid x)$ that is the Gaussian mean squared error between $y$ and the denoised $x$. The motivation for this is that it is easy to evaluate and the weights will telescope to cancel. After this telescoping, we will be left with a sequence of Markov transitions in terms of the TDS proposals and backward L-kernels. To convert this sequence of backward kernels into forward processes, we use equation 28 and again use the assumption that $\tilde{p}(y \mid x) \approx p(x \mid y)$ to give us a non-disjoint Markov chain which targets the distribution of interest.

---

**Algorithm 1** SFK with HMC kernel: refinement at every noise level.

---

**Require:** observation $y$, particles $P$, diffusion steps $T$, HMC iters $K$, leapfrog steps $L$, guidance schedule $\{\lambda_t\}$
 1: Sample $x_T^{(i)} \sim \mathcal{N}(0, I)$, set $\log w^{(i)} \leftarrow 0$ for $i = 1, \ldots, P$
 2: **for** $t = T - 1, \ldots, 0$ **do**
 3:   Propose $x_t^{(i)} \sim \tilde{p}(x_t \mid x_{t+1}^{(i)}, y)$ using Euler-Maruyama and update $\log w^{(i)}$ via equation 13
 4:   $\{x_t^{(i)}\}, \{\log w^{(i)}\} \leftarrow \text{HMCBLOCK}\big(\{x_t^{(i)}\}, \{\log w^{(i)}\}; t, y, \lambda_t, K, L\big)$
 5: **end for**
 6: **return** $\{x_0^{(i)}, \log w^{(i)}\}_{i=1}^P$

---

**Algorithm 2** HMCBLOCK: $K$ uHMC iterations at a fixed noise level.

---

**Require:** particles $\{x^{(i)}\}$, log-weights $\{\log w^{(i)}\}$, noise level $t$, observation $y$, guidance $\lambda$, iters $K$, leapfrog steps $L$, ESS threshold $\tau$
 1: Set $M \leftarrow \beta_t$, $h \leftarrow c \cdot \beta_t$                 ▷ mass and step size
 2: **if** $\text{Ess}(\log w) < \tau P$ **then**
 3:   Resample $\{x^{(i)}\}$ with weights $\propto \exp(\log w^{(i)})$;   $\log w^{(i)} \leftarrow -\log P$
 4: **end if**
 5: Compute $\log \tilde{p}(y \mid x^{(i)})$ for each $i$
 6: **for** $s$-steps **do**
 7:   Sample $v^{(i)} \sim \mathcal{N}(0, M)$
 8:   $(x'^{(i)}, v'^{(i)}) \leftarrow \text{LEAPFROG}_L(x^{(i)}, v^{(i)}; t, y, \lambda)$
 9:   Compute $\log \tilde{p}(y \mid x'^{(i)})$
10:   $\log w^{(i)} \leftarrow \log w^{(i)} + \big[\log \tilde{p}(y \mid x'^{(i)}) - \log \tilde{p}(y \mid x^{(i)})\big] + \big[\log \mathcal{N}(-v'^{(i)}; 0, M) - \log \mathcal{N}(v^{(i)}; 0, M)\big]$
11:   $x^{(i)} \leftarrow x'^{(i)}$
12:   **if** $\text{Ess}(\log w) < \tau P$ **then**
13:    Resample $\{x^{(i)}\}$ with weights $\propto \exp(\log w^{(i)})$;   $\log w^{(i)} \leftarrow -\log P$
14:   **end if**
15: **end for**
16: **return** $\{x^{(i)}\}, \{\log w^{(i)}\}$

---

**Remark (other proposals).** We adopt uHMC throughout as our intermediate target distribution refinement strategy, but Lemma 1 accommodates any proposal–backward-kernel pair admitting an analogous change-of-variables identity. Adaptive variants such as the No-U-Turn Sampler (Hoffman & Gelman, 2014) could be substituted with appropriate modifications.

### 3.3 The rSFK–uHMC sampler

We now instantiate the rSFK formulation of 3.1 and 3.1.1 with the uHMC kernel of §3.2. We give pseudocode for both the full version (corresponding to SFK with refinement at every noise level) and the reduced version (rSFK with $K_{t_\star} = K$ and $K_t = 0$ otherwise) that we run in practice. The remainder of the section documents the implementation choices – mass matrix, step size, momentum reweighting, resampling, final denoise –needed to make either variant work in image-scale state spaces.

To clarify on the choices of proposals, when we sample between timesteps, we choose the Euler-Maruyama proposal (as used in (Wu et al., 2023)) and the uHMC proposal when sampling within timesteps.

#### 3.3.1 Pseudocode

The HMC refinement at a single noise level is encapsulated as a subroutine HMCBLOCK (Algorithm 2), which iterates $K$ uHMC steps with adaptive resampling on the effective sample size. Algorithms 1 and 3 differ only in how often this subroutine is called: every reverse step in the full version, once at $t_\star$ in the reduced version.

---

**Algorithm 3** rSFK with HMC kernel at $t_\star$: refinement at a single noise level.

---

**Require:** observation $y$, particles $P$, diffusion steps $T$, refinement level $t_\star$, HMC iters $K$, leapfrog steps $L$, guidance $\lambda$

1: Sample $x_T^{(i)} \sim \mathcal{N}(0, I)$, set $\log w^{(i)} \leftarrow 0$
2: **for** $t = T - 1, \ldots, t_\star$ **do**
3:     Propose $x_t^{(i)} \sim \tilde{p}(x_t \mid x_{t+1}^{(i)}, y)$ using Euler-Maruyama and update $\log w^{(i)}$ via equation 13
4: **end for**
5: Resample $\{x_{t_\star}^{(i)}\}$ with weights $\propto \exp(\log w^{(i)})$;   $\log w^{(i)} \leftarrow -\log P$
6: $\{x_{t_\star}^{(i)}\}, \{\log w^{(i)}\} \leftarrow \text{HMCBLOCK}\big(\{x_{t_\star}^{(i)}\}, \{\log w^{(i)}\}; t_\star, y, \lambda, K, L\big)$
7: $x_0^{(i)} \leftarrow \big(x_{t_\star}^{(i)} - \sqrt{1 - \bar{\alpha}_{t_\star}}\, \epsilon_\theta(x_{t_\star}^{(i)}, t_\star)\big)/\sqrt{\bar{\alpha}_{t_\star}}$             $\triangleright$ Tweedie denoise to $x_0$
8: **return** $\{x_0^{(i)}, \log w^{(i)}\}_{i=1}^P$

---

### 3.3.2   Why a single $\lambda$ rather than a schedule

Beyond the computational savings already noted in §3.1.1, instantiating the rSFK reduction with the HMC kernel has a statistical motivation specific to the choice of guidance scale. The scalar $\lambda$ controls how aggressively the leapfrog dynamics chase the twisted-likelihood gradient, and the appropriate value depends strongly on the noise level. At high noise, $\tilde{p}(y \mid x_t)$ is a poor proxy for $p(y \mid x_t)$ and its gradient is correspondingly noisy; pushing it hard with large $\lambda$ amplifies this noise. At low noise the twisted likelihood is closer to exact and stronger guidance is justified. Algorithm 1 therefore requires either a flat $\lambda$ – a poor compromise – or a tuned schedule $\{\lambda_t\}$ across hundreds of noise levels, with all the experimental overhead that entails. Concentrating refinement at a single $t_\star$ replaces this schedule with a single scalar, leaving the algorithm with a small fixed search space: $\lambda$, $K$, $L$, and $t_\star$.

### 3.3.3   Choosing $t_\star$

The reduction in Algorithm 3 concentrates the refinement budget at one noise level, which raises the question of where to place it. Two arguments favour low-noise placement.

The twisted likelihood $\tilde{p}(y \mid x_t)$ is constructed by plugging Tweedie's $x_0$-prediction into the exact likelihood $p(y \mid x_0)$. The quality of this approximation improves monotonically as $t \to 0$, because the $x_0$-prediction becomes more accurate. The unconditional score $s_\theta(\cdot, t)$ that drives the leapfrog dynamics is also most informative at low noise. Concentrating refinement at a low-noise $t_\star$ therefore allocates the HMC budget to the noise level where both the guidance signal and the underlying score are most reliable.

In our experiments we find that placing $t_\star$ in the final few percent of the reverse process recovers the bulk of the gain that Algorithm 1 achieves at a fraction of the compute (see §4, Table 1).

### 3.3.4   Implementation details

Both Algorithm 1 and Algorithm 3 require a number of implementation choices that we briefly document here. We group them into choices that affect the dynamics (mass and step size, momentum reweighting) and choices that are purely algorithmic (resampling, final denoise, task-specific post-processing).

**Mass matrix and step size.** We use a scalar mass $M = \beta_t$ and step size $h = c\beta_t$ at noise level $t$, where $c$ is a fixed constant across noise levels. Tying both quantities to $\beta_t$ matches the natural scale of the diffusion process: the reverse transition $p(x_t \mid x_{t+1})$ has variance proportional to $\beta_t$, so the leapfrog dynamics propagate particles a comparable distance per step. The constant $c$ is the only genuine step-size hyperparameter and is chosen by coarse grid search.

**Stabilising the momentum log-density ratio.** The change-of-variables factor in Lemma 1 is the joint Gaussian log-density ratio $\log \mathcal{N}(-v'; 0, M) - \log \mathcal{N}(v; 0, M)$, which sums over all $d = C \times H \times W$ dimensions of the state space. In image-scale settings this sum has very high variance across particles, and substituting it directly into the SMC weight causes immediate weight collapse in our experiments. We therefore replace

---

the sum with the per-dimension mean,

$$\frac{1}{d}\sum_{j=1}^{d}\big[\log\mathcal{N}(-v'_j;0,M) - \log\mathcal{N}(v_j;0,M)\big], \tag{43}$$

which preserves the sign of the correction but scales its magnitude by $1/d$. This is a deliberate deviation from Lemma 1: the implicit SMC target is no longer exactly $\tilde{p}(y \mid x)$, and the incremental weight is no longer a strict change-of-variables factor. We adopt the substitution because, empirically, it maintains stable effective sample size while preserving the qualitative effect of the momentum correction.

When taking the mean of the velocity evaluations as described in equation 43, the incremental weight update during the HMC refinement stage becomes:

$$\frac{p(x' \mid y)}{p(x \mid y)} \cdot \left(\frac{L(x \mid x', y)}{\tilde{p}(x' \mid x, y)}\right)^{\frac{1}{D}}, \tag{44}$$

where $D$ is the number of pixels and we assume a single refinement iteration. However, taking only the mean log probability overthe velocities alters the targte. Therefore we can instead take the mean of the twisted functions as well:

$$\left(\frac{p(x' \mid y)}{p(x \mid y)} \cdot \frac{L(x \mid x', y)}{\tilde{p}(x' \mid x, y)}\right)^{\frac{1}{D}}. \tag{45}$$

Applying detailed balance, we can express $L$ in terms of the forward proposal:

$$\left(\frac{p(x' \mid y)}{p(x \mid y)} \cdot \frac{\tilde{p}(x' \mid x, y)}{\tilde{p}(x' \mid x, y)} \frac{p(x \mid y)}{p(x' \mid y)}\right)^{\frac{1}{D}} \tag{46}$$

$$= \left(\frac{p(x' \mid y)}{p(x \mid y)} \cdot \frac{p(x \mid y)}{p(x' \mid y)}\right)^{\frac{1}{D}} = 1. \tag{47}$$

**Resampling within HmcBlock.** The HMCBLOCK subroutine resamples particles whenever the effective sample size drops below $\tau P$ for some threshold $\tau \in (0, 1]$ (we use $\tau = 0.5$). This is standard practice in SMC and is needed because the per-iteration weight ratio $\tilde{p}(y \mid x')/\tilde{p}(y \mid x)$ can vary substantially across particles when guidance is strong.

**Forced resampling at $t_\star$.** In Algorithm 3 we resample particles deterministically at the start of the HMC phase, regardless of effective sample size. This gives the refinement block a uniformly-weighted starting population and prevents TDS weight imbalances accumulated upstream from dominating the post-refinement weights.

**Final denoise via Tweedie.** The closing step of Algorithm 3 replaces the trailing reverse iterations with a single deterministic Tweedie projection $x_0 = (x_{t_\star} - \sqrt{1 - \bar{\alpha}_{t_\star}}\,\epsilon_\theta)/\sqrt{\bar{\alpha}_{t_\star}}$. This is a deliberate shortcut: at the low noise levels where $t_\star$ is chosen, the Tweedie estimate is close to a full reverse-process completion, and avoiding the trailing stochastic steps gives a cleaner final sample with no further weight updates. Continuing the reverse process from $t_\star$ to $t = 0$ is also valid.

## 4 Experiments

We evaluate rSFK-uHMC against TDS across four benchmark image datasets: MNIST (Lecun et al., 1998), Flowers (Nilsback & Zisserman, 2006), Smithsonian Butterflies, and (Liu et al., 2015), and four conditional sampling tasks are considered. For inpainting, two masking strategies are evaluated: a centre box mask occluding the central 50% of the image, and a random pixel mask removing 70% of pixels uniformly at random. For super-resolution, images are downsampled by a factor of $2\times$ for resolutions up to $64 \times 64$ and $4\times$ for higher resolutions, using bicubic interpolation. For deblurring, a Gaussian blur kernel is applied with

Table 1: Full results comparing TDS and rSFK-uHMC across all datasets and tasks. **Bold** indicates the best value per metric per row.

| Dataset | Task | TDS | | | rSFK-uHMC | | |
|---|---|---|---|---|---|---|---|
| | | PSNR ↑ | SSIM ↑ | LPIPS ↓ | PSNR ↑ | SSIM ↑ | LPIPS ↓ |
| MNIST | Center Mask | 14.03 | 0.755 | 0.108 | **14.22** | **0.758** | **0.105** |
| | Random Mask | 17.25 | 0.822 | 0.081 | **18.48** | **0.852** | **0.063** |
| | Super-Res. | 24.87 | 0.970 | 0.018 | **28.79** | **0.988** | **0.009** |
| | Deblurring | 17.59 | 0.847 | 0.067 | **25.21** | **0.970** | **0.020** |
| Flowers | Center Mask | 20.25 | 0.788 | **0.107** | **20.27** | **0.794** | 0.108 |
| | Random Mask | 22.93 | 0.725 | 0.082 | **24.41** | **0.801** | **0.044** |
| | Super-Res. | 22.59 | 0.660 | 0.063 | **23.16** | **0.696** | **0.060** |
| | Deblurring | 20.86 | 0.544 | 0.100 | **22.52** | **0.652** | **0.083** |
| Butterflies | Center Mask | 21.81 | 0.814 | **0.079** | **21.91** | **0.820** | 0.080 |
| | Random Mask | 24.46 | 0.797 | 0.077 | **26.17** | **0.861** | **0.038** |
| | Super-Res. | 23.75 | 0.725 | 0.069 | **24.66** | **0.768** | **0.061** |
| | Deblurring | 21.51 | 0.620 | 0.107 | **23.40** | **0.715** | **0.082** |
| CelebA | Center Mask | 25.54 | 0.870 | **0.055** | **25.61** | **0.877** | 0.057 |
| | Random Mask | 32.90 | 0.900 | 0.060 | **34.80** | **0.936** | **0.035** |
| | Super-Res. | 31.53 | 0.851 | **0.058** | **32.45** | **0.881** | 0.068 |
| | Deblurring | 28.97 | 0.778 | **0.088** | **30.22** | **0.827** | 0.094 |

kernel size and standard deviation scaled to the image resolution; specifically, kernel sizes of $9, 17, 31, 61$ and blur standard deviations of $1.5, 2.0, 2.5, 3.0$ are used for resolutions $32, 64, 128, 256$ respectively.

For each dataset, 100 images are sampled from a pre-trained DDPM model hosted on HuggingFace, and all results are averaged over these 100 images. Using model-sampled images rather than held-out test images ensures that the prior is well-matched to the data, providing a clean testbed for isolating the effect of the sampler.

We report three metrics per method. Peak signal-to-noise ratio (PSNR) and structural similarity index measure (SSIM) (Horé & Ziou, 2010) are standard full-reference quality metrics, with higher values indicating closer reconstruction to the original. Learned perceptual image patch similarity (LPIPS) (Zhang et al., 2018) measures perceptual similarity using deep features, where lower values are preferable.

Drawing from the hyperparameter choices of Du et al. (2023), rSFK-uHMC uses $L = 3$ leapfrog steps, a step size of $0.3\beta_t$ and a mass matrix of $\beta_t I$ with $s_t = 20$ refinement iterations. For the MNIST dataset, 100 denoising steps $\mathcal{K}$ are used, with Butterflies and Flowers using 500 and CelebA using 1000. The HMC refinement was applied at the 95% iteration (95 for MNIST, 475 for Butterflies and Flowers, 950 for CelebA) and the number of particles $N$ used during the sample process was 16, 8, 8 and 4 respectively due to memory restrictions. The gradient-guidance scale was set to $\gamma = 1000, 500, 500, 100$ respectively.

## 5 Discussion

Table 1 presents quantitative results across all conditional sampling experiments and datasets, with Figure 2 showing example reconstructions. Across nearly all tasks and metrics, rSFK-uHMC outperforms TDS, with the exception of the LPIPS metric on CelebA. This is not necessarily contradictory; LPIPS measures perceptual similarity using deep neural network features and is known to sometimes disagree with pixel-level metrics such as PSNR and SSIM (Zhang et al., 2018). In particular, PSNR and SSIM reward pixel-level

fidelity, whereas LPIPS better captures high-level perceptual structure. It is possible that TDS produces reconstructions that are perceptually plausible but pixel-level inaccurate, whilst rSFK-uHMC recovers finer pixel-level detail at the cost of a marginal perceptual penalty.

The performance gains are somewhat less pronounced for the center mask inpainting task relative to other conditional experiments. We hypothesise that this is because the center mask creates a large contiguous region of missing information that is spatially disconnected from the surrounding observed pixels, making it harder for any sampler to confidently recover fine-grained structure in the occluded region.

We also note that there is a modest computational overhead is incurred by the uHMC refinement stage. This overhead is most pronounced for lower-resolution datasets: for MNIST, the base diffusion trajectory uses 100 steps, and the uHMC phase adds a further 60 gradient evaluations (20 SMC iterations $\times$ $L = 3$ leapfrog steps) at the 95th timestep, bringing the total to 160 gradient steps, a 60% increase. For Flowers and Butterflies, the base trajectory uses 500 steps, with uHMC contributing an additional 60, for a total of 560 (a 12% increase). For CelebA, the equivalent figures are 1000 and 1060 steps respectively (a 6% increase). The relative runtime penalty therefore diminishes substantially as the base number of sampling steps grows.

Qualitatively, the reconstructions in Figure 2 appear visually similar across both methods, consistent with both achieving reasonable image quality. Differences are most apparent in the random pixel mask and deblurring tasks, where rSFK-uHMC reconstructions appear slightly sharper, though the quantitative metrics in Table 1 more reliably capture the performance gap.

A further practical advantage of rSFK-uHMC over TDS-HMC is the ease of tuning the gradient guidance scale. In TDS-HMC, the optimal guidance scale may need to be annealed across the full diffusion trajectory, as the relative contribution of the score function and likelihood gradient varies across timesteps, requiring a carefully chosen schedule. In rSFK-uHMC, the uHMC phase operates at a single timestep near the end of the trajectory, meaning the guidance scale need only be tuned for that one point. This substantially reduces the hyperparameter search space compared to specifying a full annealing schedule.

### 5.0.1 Similar NFE

A potential criticism of our proposed method is that rSFK-uHMC requires a greater number of function evaluations (NFEs) than standard TDS, which may raise concerns about computational fairness in our comparisons. To address this directly, Table 2 presents a controlled comparison in which both methods are allocated a comparable NFE budget for the Random-Mask and Super-resolution tasks on the MNIST, Flowers and Butterflies dataset. Specifically, since rSFK-uHMC uses approximately 60 more evaluations per sample than TDS, we augment TDS with an additional 75 NFEs per sample to ensure a roughly equivalent computational budget. Despite this adjustment, rSFK-uHMC continues to outperform TDS across the reported metrics, demonstrating that its performance gains are not merely a consequence of greater computational expenditure, but reflect a genuine algorithmic advantage.

Table 2: Comparison of TDS and rSFK-uHMC at similar NFEs and wall-clock time. **Bold** indicates the best value per metric per row.

| Dataset | Task | TDS | | rSFK-uHMC | |
|---|---|---|---|---|---|
| | | PSNR ↑ | SSIM ↑ | PSNR ↑ | SSIM ↑ |
| MNIST $\mathcal{K}_{TDS} = 170, \mathcal{K}_{rSFK} = 100$ | Random Mask | 17.90 | 0.835 | **18.48** | **0.852** |
| | Super-Res. | 25.51 | 0.976 | **28.79** | **0.988** |
| Flowers $\mathcal{K}_{TDS} = 575, \mathcal{K}_{rSFK} = 500$ | Random Mask | 22.66 | 0.710 | **24.41** | **0.801** |
| | Super-Res. | 22.47 | 0.656 | **23.16** | **0.696** |
| Butterflies $\mathcal{K}_{TDS} = 575, \mathcal{K}_{rSFK} = 500$ | Random Mask | 24.49 | 0.792 | **26.17** | **0.861** |
| | Super-Res. | 23.80 | 0.726 | **24.66** | **0.768** |

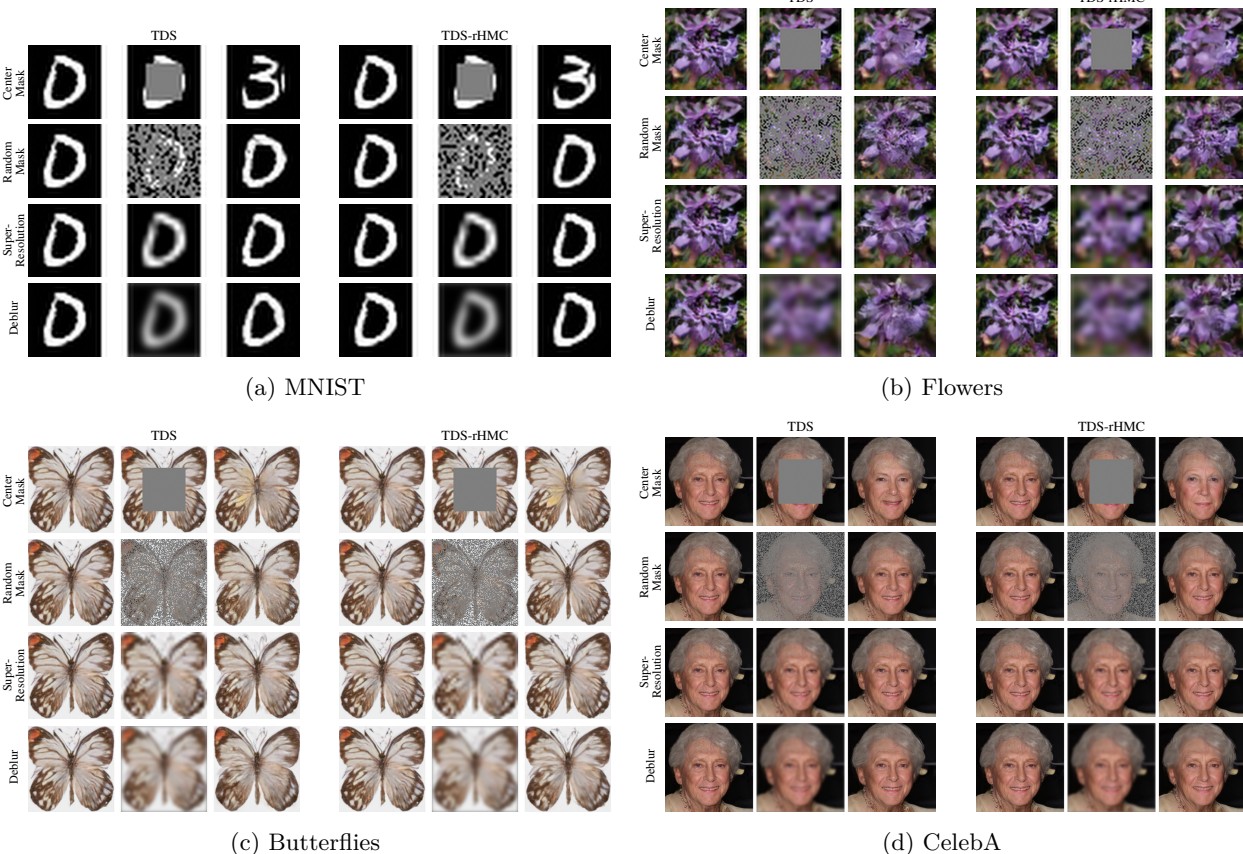

Figure 2: Qualitative reconstruction comparisons across all datasets. Each sub-figure shows TDS (left column) vs. rSFK-uHMC (right column), with rows corresponding to different inverse problem tasks.

## 5.1 Stopping Point

We also provide ablation studies for trying to find the optimal early stopping time for the rSFK-uHMC method on the MNIST dataset and Flowers dataset. We see that after a certain point, the metrics stop increasing and start to decrease. If we stop too late, the samples are bound by the estimates made by the TDS implementation up to this point and therefore it is hard to improve upon the images already created. The sample diversity is also degenerate as the noise level induces a peaked likelihood which means samples cannot differ too much from one another without the effective sample size decreasing. If we stop too early then the TDS sampler has not been able to denoise the images enough and the uHMC componenet struggles to refine the images as the starting distribution estimation is too crude. This could potentially be fixed by more uHMC iterations. Further study upon this is required.

## 5.2 Limitations and Further Work

One potential failure mode is bad calibration of the gradient guidance in the uHMC step. We found during preliminary results that simply setting the guidance value to an arbitrarily high value led to degredation in the image reconstructions, often with images being oversaturated and speckled-like with brighter pixels. We also recognise that the effect of uHMC may be mitigated by a TDS implementation which has a calibrated guidance parameter schedule.

We found in our work that stopping early and using rSFK increased particle diversity as well, which is a common issue in SMC-based methods for conditional sampling form diffusion priors. Due to the sampler stoppage before very small variance values, the degeneracy of the sampling method is capped. However

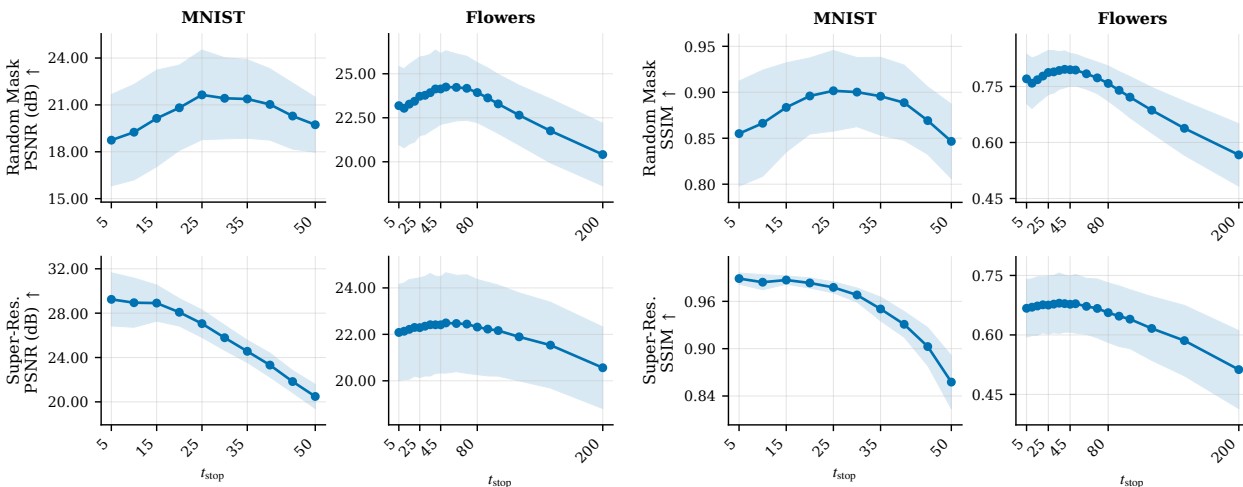

Figure 3: Effect of $t_{\text{stop}}$ on PSNR (left) and SSIM (right) across datasets and tasks. Shaded bands show $\pm 1$ standard deviation over 30 images.

for the samples to converge at the final target, we must evaluate $\tilde{p}(y \mid x_0^0)$ which often has a very small corresponding variance value, causing particle diversity to collapse after this final evaluation. Investigating an alternative final evaluation which would allow for particle diversity would allow SMC-based methods to potentially overcome the degeneracy issue and would be a very fruitful area of research for future work.

Recent advances in diffusion model sampling have focused on improved integrators (Karras et al., 2022; Xue et al., 2023; Lu et al., 2022), which require fewer NFEs to produce high-fidelity images. Most of these methods cannot be used directly as proposals within rSFK, as they do not admit a tractable, point-evaluable density. Nevertheless, a promising direction would be to combine such integrators with rSFK in the final iterations, using the integrator to propagate particles efficiently between timesteps and rSFK to refine samples in a similar fashion as the research conducted in this paper.

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
