# OpenReview forum: "Stacked Feynman-Kac: A Generalised Method for Within-Timestep Sampling of Intermediate Distributions for Diffusion Models"
_TMLR — Under review for TMLR_

### Review · Reviewer_z4K1 · 2026-07-01

**Summary Of Contributions:**

The paper proposes Stacked Feynman-Kac (SFK), a framework for improving SMC-based conditional sampling in diffusion models by adding within-timestep refinement steps that target intermediate conditional marginals $p(x_t|y)$. The authors instantiate this idea with an unadjusted HMC-style proposal and SMC reweighting, and introduce a reduced version, rSFK-uHMC, that applies this refinement only near the end of the reverse diffusion trajectory to reduce compute. Experiments on several image inverse problems show improved PSNR and SSIM over TDS, with moderate additional NFE cost.

The main strength is that the paper identifies a meaningful limitation of existing diffusion-SMC samplers: particles are propagated between timesteps but may insufficiently explore the conditional marginal at each timestep. The proposed within-timestep refinement is conceptually interesting and appears complementary to SMC-based conditional sampling.

However, the empirical support is currently incomplete. The comparisons are mostly limited to TDS, without comparisons to other within-timestep corrector or MCMC-style methods. While the contribution is orthogonal to the type of SMC, the experiments are only done on a single sampler, TDS. The evaluation also focuses on distortion metrics such as PSNR and SSIM, which may not fully reflect sample quality under the perception-distortion trade-off. Finally, the practical implementation includes approximations that weaken the asymptotic-exactness story, and the claimed generality of the SFK framework is not tested across different SMC samplers.

**Audience:**

Yes

**Audience Explanation:**

Yes. Conditional sampling with diffusion priors is an active area, and SMC-based approaches are of interest because they provide a principled alternative to heuristic posterior-score approximations. The idea of using SMC not only across diffusion timesteps but also for within-timestep refinement is conceptually interesting. If the empirical concerns are addressed, the paper would likely be useful to researchers working on diffusion posterior sampling, SMC for generative models, and inverse problems.

**Broader Impact Concerns:**

I do not see major broader-impact concerns beyond those generally associated with diffusion-based image restoration and conditional generation. The method could potentially be used for image manipulation or reconstruction in sensitive settings, but this is not substantially different from existing diffusion restoration methods. A brief broader-impact statement acknowledging such uses would be sufficient.

**Claims And Evidence:**

No

**Claims Explanation:**

The paper provides promising evidence that rSFK-uHMC improves reconstruction metrics over TDS, including a useful similar-NFE comparison. However, I do not think the current evidence is sufficient to fully support the paper’s main empirical and conceptual claims.

First, the paper’s main motivation is that within-timestep refinement better respects intermediate conditional marginals, but the experiments do not compare against other natural methods that also aim to improve within-timestep sampling, such as Langevin correctors, MALA-style correctors, or other MCMC/corrector methods. This makes it difficult to isolate whether the gain comes from the proposed SFK formulation specifically, from using additional likelihood-gradient updates, or from simply changing the sampling regime. More, while the contribution is orthogonal to the type of SMC, the experiments are only done on a single sampler, TDS.

Second, the reported improvements are primarily in PSNR and SSIM. These are distortion-oriented metrics, and improvements in them may correspond to moving along the perception-distortion trade-off rather than improving generative sample quality. LPIPS is reported, but it is not sufficient by itself. The paper should also report FID, and ideally additional distributional/diversity metrics, to clarify whether the proposed sampler improves posterior sample quality or mainly produces more distortion-optimal reconstructions.

Third, the implementation deviates from the exact theoretical formulation, especially in the stabilized momentum-ratio treatment. This practical modification appears important for making the method work, but it also weakens the exactness claim. The paper should be clearer about which claims apply to the idealized algorithm and which apply to the implemented rSFK-uHMC sampler.

**Requested Changes:**

1. **Add comparisons to other within-timestep correction methods**: The paper should compare against methods that also attempt to improve intermediate conditional sampling, such as Langevin correctors, MALA-style correctors, or other MCMC/corrector variants. Without the comparisons, it is hard to tell whether the gains are due to the proposed SFK formulation or simply due to adding likelihood-gradient refinement steps.

2. **Test complementarity with different SMC samplers**: The proposed framework is complementary to other SMC samplers, but the experiments only instantiate it with TDS. The authors should test rSFK with additional SMC samplers.

3. **Report FID scores**: Since improved PSNR/SSIM may simply indicate a different point on the perception-distortion trade-off, the authors should report FID, preferably computed with enough samples to be meaningful.

4. **Improve discussion on failure modes**: The paper mentions that overly strong guidance can lead to oversaturated or speckled samples. More systematic analysis of this failure mode would help readers understand when the method is likely to work or fail.

---

### Review · Reviewer_X4m4 · 2026-07-06

**Summary Of Contributions:**

The paper proposes an algorithm for conditional sampling from an unconditional diffusion model. The algorithm is based on the framework of Feynman-Kac (FK) algorithms and mainly builds on the specific instantiation called twisted diffusion sampler (TDS).

The main idea is to include an HMC correction in each step of the TDS algorithm. This requires in particular to maintain an ELBO-like particle weight during the leapfrog steps of HMC via change-of-variables formulas. In order to keep the computational cost on a tractable level, the authors propose to use the HMC correction only in the latest steps. The resulting algorithm is called (reduced) Stacked Feynman-Kac (SFK). The approach is valid and up to some smaller issues (see other fields) sound.

The performance of the reduced SFK algorithm with HMC is evaluated on some examples. However, the precise setup is very simplistic, not always clear and doesn't show any standard benchmarks, so it is a bit hard to judge how well the algorithm is really performing.

**Audience:**

Yes

**Audience Explanation:**

The paper proposes a conditional sampling algorithm from unconditional diffusion model. This is clearly a topic of interest for many TMLR readers.

**Claims And Evidence:**

Yes

**Claims Explanation:**

The idea of including HMC correction steps in TDS is generally sound. Also the computation of the change of variables formula and the consistency proof in Thm 1 looks sound, even though I did not check every computation. There is a couple of smaller issues (missing assumptions and clarity problems) included in the "requested changes" part.

The experimental part is currently (in my opinion) insufficient to support the claim of "superior reconstruction performance with minimal additional computation" and requires significantly more efforts. Please refer to the "requested changes" part for a detailed list.

**Requested Changes:**

Methodological part:

- The change-of-variables formula in eqt (21) and Lemma 1 are not fully correct. They require injectivity of the leap frog map $f_{LF}$ (otherwise more general area formulas have to be used, which is quite complicated). For the presented case of $L=1$ this is quite clear as $f_{LF}(x,v)=hv + g(x)$ for some function g. However, for $L>1$ this requires additional considerations. I guess one could derive a sufficient condition by assuming that $\log\pi$ is Lipschitz and $h$ is small enough (probably the upper bound is proportional to $1/\mathrm{Lip}(\log\pi)$).

- Literature: A similar way for including MCMC sampling steps in generative models was proposed under the name "Stochastic Normalizing Flows". This line of works should be considered in the related works part.

- Thm 1 shows that in the "no-errors setting" SFK produces the correct distribution. However, the same result is was shown for TDS. So if both algorithms generate the same distribution, why is SFK better? In other words, the authors should explain where errors in TDS arise and how this is addressed/improved by SFK.

Experiments:

- Posterior sampling vs point reconstruction: The authors propose an algorithm for sampling the posterior distribution. However they don't evaluate at all how well this is done. Instead they use error metrics like PSNR/SSIM/LPIPS which are tailored for point reconstructions (like MAP/MMSE estimates). This might be ok for identifiable problems like deblurring, but does not make any sense for the box-inpainting problem (and one can also see in the MNIST example in Fig 2a that the 3 is a totally fine reconstruction even though any of these metrics it will be treated as really bad). I am clear that evaluating the quality of posterior samplers is not straightforward. But there are some experiments which can be done, two suggestions: 1. Train a diffusion model on a 2D-Gaussian mixture, then the posterior can be analytically derived. Then sample with your method from the posterior and compare in some metric (MMD/Wasserstein or similar). 2. Do class-conditional sampling for MNIST and compare the FID between the generated samples and the training samples in a certain class.

- The authors write nothing about the noise (or noise level) from the setup in the experiments (the $\epsilon$ in eqt (1)). From looking at the images it is not visible whether the authors apply noise at all. However, the analysis from before strictly requires the presence of noise on the observation $y$ (otherwise the conditional densities become singular and everything becomes more complicated).

- Evaluation metrics: For inverse problems it is crucial to ensure that the generated images remain faithful to the observation data. Therefore I would strongly recommend to evaluate the data consistency $\frac{\|\mathcal{A}(\hat x)-y\|^2}{\mathbb{E}[\|\epsilon\|^2]}$ for the reconstruction $\hat x$. If the reconstruction is faithful to the data, this value should be between zero and one. Values significantly larger than one indicate a serious problem.

- For the overall positioning of the method it would be good to add additional baselines in the examples (such as DPS or DiffPIR). Also, for the deblurring example, the authors should report the error metrics for the pseudo-inverse. This shouldn't be too much work as well since there are many openly available implementations of these baselines for very similar problems.

- For papers with mostly empirical results (like this one), I find it crucial that the results are reproducible. To ensure reproducibility the authors have two options: The easier one is to make the code publicly available (e.g. as supplementary material). Alternatively, the authors can also include a precise description of all used architectures and hyperparameters in the paper.

- Since the paper comments a lot about the runtime, the authors should report the evaluation time and memory requirements for at least one of the examples and compare it to the baselines.

- Overall, the experiments are done in a super-toyish setup: images sampled from the exact diffusion model, no (unclear? Gaussian?) noise model, simplistic computer-vision-like forward operators, very homogeneous datasets and so on. It is really hard to say how the proposed method would perform in a more realistic setting. To a certain extent it is ok to use a simplified setup for the initial testing of such a method. However, in my opinion, in particular the clear statement of the noise model, an investigation of the generalization to test images which are not exactly generated from the diffusion model are crucial. Also some more diverse dataset would be good. Finally, the limited evaluation should be declared in the limitations.

Minor:

- Notation: $\nu_T$ (eqt (11) and later) is not a good notation, since it takes the full trajectory as input. A better notation would be $\nu_{0:T}$.

- Fig 2 would benefit from some zoom-in parts (e.g. with tikz spy). Otherwise it is hard to see anything in the butterfly/celeba/Flowers images.

---

### Review · Reviewer_HL4k · 2026-07-15

**Summary Of Contributions:**

**Summary**

The paper addresses conditional sampling from diffusion priors for inverse problems via SMC-based algorithm. Its observation is that existing SMC samplers such as TDS propagate particles only across timesteps and never refine the intermediate target within a fixed noise level. The authors propose Stacked Feynman-Kac (SFK), a Feynman-Kac formulation that inserts K applications of a Markov kernel that leaves intermediate timestep distribution invariant. They instantiate the kernel with an uHMC using Leapfrog proposal, replacing the Metropolis-Hastings accept/reject with an importance weight. A reduced variant (rSFK) applies this refinement at a certain timestep interval to reduce computation cost. Experiments on image benchmarks show that rSFK + uHMC improves PSNR or SSIM comparing with TDS baseline.

**Strengths**

The motivation about within-timestep refinement is clean and the FK formalization is a reasonable generalization of TDS.
The rSFK reduction is well-motivated and the NFE-matched comparison (Table 2) is a good-faith attempt at fairness.

**Weaknesses**

The central exactness argument appears to collapse (Eq. 45–47 drive the within-timestep weight to 1, which is inconsistent with the paper's own claim); the only baseline is TDS, with no comparison to non-particle guidance or, critically, to a Langevin corrector; and notation is inconsistent to the point of obstructing verification of the proofs.

**Audience:**

Yes

**Audience Explanation:**

Conditional sampling from diffusion priors via SMC is an active topic, and the specific idea of within-timestep refinement inside a Feynman-Kac formulation is an useful direction that would interest the diffusion-sampling and Monte-Carlo audiences.

**Broader Impact Concerns:**

No Broader Impact Statement is present.

**Claims And Evidence:**

No

**Claims Explanation:**

The paper's headline claim is that SFK provides principled within-timestep refinement with the uHMC dynamics corrected by the SMC importance weight (Eq. 24/40) rather than by using the Metropolis-Hastings acceptance ratio. Two issues undermine this.


First, the weight derivation in Sec.3.3.4 appears inconsistent. To avoid weight collapse in image-scale spaces, the momentum log-density ratio is replaced by its per-dimension mean (Eq. 43). Then Eq. 45–47 take the mean of the twisted factors as well and, via detailed balance, the incremental weight reduces to exactly 1. A within-timestep weight identically equal to 1 means refinement contributes no importance correction at all — precisely the uncorrected regime the paper claims. This also contradicts the implementation in Algorithm 2 (line 10). So it is unclear which target the method actually samples.


Second, the empirical support is thin relative to the claim of a general improvement to SMC diffusion sampling. The sole baseline is TDS, and there is no comparison to non-particle guidance - DPS (Chung et al., 2023), ΠGDM (Song et al., 2023), DDRM (Kawar et al., 2022), DDNM (Wang et al., 2023) - or to other particle/SMC methods such as FPS (Dou & Song, 2024) and MCGdiff (Cardoso et al., 2024). Most importantly for a paper whose contribution is a within-timestep corrector, there is no comparison to the alternative of a Langevin corrector applied at the same noise level. The predictor–corrector framework of Song et al. (2021b) already interleaves reverse-diffusion "predictor" steps with score-based Langevin "corrector" steps that, at a fixed noise level, run several iterations to pull particles toward the marginal $p(x_t)$, the exact same "refine within a timestep" role that SFK's HMC block plays, and one that extends to the conditional score $s_\theta(x,t) + \lambda \nabla \log \tilde p(y\mid x_t)$ with no accept/reject step. Since the deployed uHMC uses Leapfrog steps with per-step momentum refresh and a scalar mass, it is conceptually same as a guided-Langevin corrector. Also, the evaluation is restricted to low-resolution ($\leq$128).

**Requested Changes:**

1. Resolve the weight inconsistency. Clarify what the incremental within-timestep weight actually is: if Eq. 47 gives 1, state plainly what target the refinement then samples and why it still improves over no refinement; if Algorithm 2 (non-trivial weight) is what is run, correct Eq. 45–47 and re-derive the implied target.

2. Add related baselines. This is the decisive comparison for whether the HMC proposal is doing more than guided Langevin. Broaden baselines beyond TDS to at least one non-particle guidance method (e.g., DPS or ΠGDM) and, if possible, one other particle/SMC method, under matched NFE.

3. Report particle diversity or ESS trajectories. Diversity collapse is central to the SMC-based method.

4. Evaluate images at least one higher-resolution/natural-image setting, to show gains are not specific to model-sampled, low-resolution data.

5. Fix notation so the proofs are checkable. Unify $K$ or $s$​ into one symbol, and define $x^s_t$ wherever it appears.

6. Clarify the relationship to plain Langevin-based corrector sampling and to DPS explicitly, positioning against those would sharpen the novelty claim.